# IgE-Induced Mast Cell Activation Is Suppressed by Dihydromyricetin through the Inhibition of NF-κB Signaling Pathway

**DOI:** 10.3390/molecules26133877

**Published:** 2021-06-25

**Authors:** Tsong-Min Chang, Tzu-Chih Hsiao, Ting-Ya Yang, Huey-Chun Huang

**Affiliations:** 1Department of Applied Cosmetology, Hungkuang University, Taichung 433304, Taiwan; ctm@hk.edu.tw; 2Juwenlee Cosmetics Technology Center, LUO LI-FEN Group, Zhangzhou 363105, China; atropa@ms18.hinet.net; 3Department of Medical Laboratory Science and Biotechnology, College of Medicine, China Medical University, Taichung 404333, Taiwan; tyyang@mail.cmu.edu.tw

**Keywords:** dihydromyricetin, mast cells, NF-κB, tryptase, STAT5

## Abstract

Mast cells play a crucial role in the pathogenesis of type 1 allergic reactions by binding to IgE and allergen complexes and initiating the degranulation process, releasing pro-inflammatory mediators. Recently, research has focused on finding a stable and effective anti-allergy compound to prevent or treat anaphylaxis. Dihydromyricetin (DHM) is a flavonoid compound with several pharmacological properties, including free radical scavenging, antithrombotic, anticancer, and anti-inflammatory activities. In this study, we investigated the anti-allergic inflammatory effects and the underlying molecular mechanism of DHM in the DNP-IgE-sensitized human mast cell line, KU812. The cytokine levels and mast cell degranulation assays were determined by enzyme-linked immunosorbent assay (ELISA). The possible mechanism of the DHM-mediated anti-allergic signaling pathway was analyzed by western blotting. It was found that treatment with DHM suppressed the levels of inflammatory cytokines TNF-α and IL-6 in DNP-IgE-sensitized KU812 cells. The anti-allergic inflammatory properties of DHM were mediated by inhibition of NF-κB activation. In addition, DHM suppressed the phosphorylation of signal transducer and activator of transcription 5 (STAT5) and mast cell-derived tryptase production. Our study shows that DHM could mitigate mast cell activation in allergic diseases.

## 1. Introduction

The leaves of *Ampelopsis grossedentata* are a type of caffeine-free tea that is widely used to promote health. This plant is rich in polyphenols and flavonoids, and its main compound is ampelopsin, also called dihydromyricetin (DHM). DHM is a bioactive flavonoid compound that exhibits many pharmacological effects, such as anti-inflammatory, hepatoprotective, anti-carcinogenic, antioxidative, and anti-melanogenesis properties [1,2,3,4]. Previous studies have shown that flavonoids reduce allergic reactions [5]. Due to its beneficial activity, DHM has been extensively studied for structural identification, content determination, and pharmacological effects [1]. To date, little has been reported regarding the precise molecular targets by which DHM regulates human mast cell function. Importantly, it has been reported that naturally occurring polyphenoloic antioxidants, such as rutin and chlorogenic acid, could modulate IgE-mediated mast cell activation [6]. The polyphenolic structure of DHM and its antioxidant activity led to our investigation of its effect on IgE-mediated mast cell activation. However, little has been reported regarding the precise molecular mechanism by which DHM inhibits human mast cell function.

There is emerging evidence that mast cells (MCs) play a central role in viral infections, systemic allergies, asthma, neuro-inflammatory diseases, and several stress disorders [7,8]. Mast cells activated in an allergen- and immunoglobulin E (IgE)-dependent manner subsequently trigger acute inflammatory reactions and promote chronic allergy progression by secreting histamine, proteases, and chemotactic factors, and engage in de novo synthesis of inflammatory cytokines. However, extensive mast cell activation leads to increased levels of inflammatory cytokines and chemokine release, which further exacerbates inflammation and increases disease severity [9]. In humans, two subsets of mast cells have been identified and characterized according to their expression of tryptase or chymase within cytoplasmic granules. The tryptase- and chymase-positive mast cell (MCTC) subtype encompasses the skin mast cells that express both tryptase and chymase, whereas the tryptase-positive mast cell (MCT) subpopulation only expresses tryptase and is predominately located in the lungs [8]. The NF-κB signaling cascade is responsible for the pro-inflammatory cytokines, TNF-α and IL-6, in activated mast cells. Triggered signaling of the Ras-Raf-Map kinase (MAPK) cascade consequently leads to the activation of a number of transcription factors required for the biological function of MCs. Moreover, signal transducers and activators of transcription 5 (STAT5) are critical for mast cell development, which increases the transcription of pro-inflammatory cytokines [10,11]. The critical contribution of STAT5 to mast cell biology was identified as the regulation of IgE-induced degranulation of mast cells [10,12]. In this study, the potential anti-allergic effects of DHM on IgE-mediated mast cell activation were investigated. We sensitized mast cells with dinitrophenol (DNP) IgE and then used DNP-human serum albumin antigen stimulation to activate the sensitized mast cells in vitro. The molecular mechanisms underlying the effects of DHM on IgE-mediated activation were also explored using human KU812 mast cells.

## 2. Results

### 2.1. Effect of DHM on the Proliferation of KU812 Cells

To determine the concentrations of DHM in our study, we sensitized KU812 cells with 100 ng/mL DNP-IgE for 16 h and stimulated with DNP-HSA for 4 h. Trypan blue assay data showed significant hyperproliferation compared to non-sensitized mast cells. The dosage effect of DHM on KU812 cells was evaluated by treatment with 10 μM and 100 μM DHM for 24 h; as observed, 100 μM DHM could more effectively inhibit IgE-induced mast cell proliferation. Treatment with 100 μM DHM for 24 h had no significant cytotoxic effect on KU812 cells (Figure 1).

### 2.2. Inhibitory Effects of DHM on the Generation of Oxidative Stress

Intracellular oxidative stress was evaluated by treating KU812 cells with DHM followed by DNP-IgE challenge. The cellular reactive oxygen species (ROS) levels were measured by flow cytometry and were dramatically right shifted in the DNP-IgE-treated cells, whereas the ROS levels were reduced in cells treated with 100 μM DHM. These results demonstrate that DHM can effectively scavenge ROS and reduce the oxidative stress induced by DNP-IgE in mast cells (Figure 2).

### 2.3. DHM Decreases DNP-IgE-Upregulated Pro-Inflammatory Cytokines and Inhibits Ige-Dependent Activation of the MAPK and NF-κB Signaling Pathways in KU812 Cells

KU812 cells were sensitized with DNP-IgE for 16 h followed by DNP-HSA stimulation for 4 h prior to DHM treatment. IgE activation results in the release of pro-inflammatory mediators, leading to the exacerbation of allergic responses. We investigated whether DHM treatments were able to modulate the release of the pro-inflammatory cytokines TNF-α and IL-6. DHM treatment significantly reduced the IgE-induced upregulation of TNF-α, in a dose-dependent manner (Figure 3A). DHM treatment significantly inhibited IL-6 cytokine levels (Figure 3B).

As the activation of MAPK and NF-κB is essential for the transcriptional regulation of allergic responses, we further investigated whether the inhibition of allergic responses by DHM is mediated through the NF-κB pathway in KU812 cells. IgE-induced activation of NF-κB was reduced by DHM treatment (Figure 3C). It has been reported that the activation of NF-κB is triggered by mitogen-activated protein kinases (MAPKs), including extracellular signal-regulated kinase (ERK), c-Jun NH2-terminal kinase (JNK), and p38 MAPK. As shown in Figure 3C, DHM suppressed the IgE-induced activation of p38 MAPK, but did not affect the phosphorylation of ERK or JNK.

### 2.4. DHM Promotes Recovery from IgE-Induced Degranulation

Upon activation, mast cells release a myriad of preformed factors, and the levels of tryptase were determined in KU812 supernatants. IgE challenge induced a significant increase in tryptase activity. The addition of 100 μM DHM reduced the IgE-induced release of tryptase (1.75 ± 0. 1 μg/mL) to (1.13 ± 0.211 μg/mL), which was nearly 64.5% of the value of the DNP-IgE group (Figure 4). These findings indicate that DHM attenuates IgE-induced mast cell degranulation.

### 2.5. DHM Inhibited IgE-Mediated Mast Cell Activation through Suppression of STAT5 Signaling Pathway

Hyperactivated STAT5 leads to the aberrant expression of pro-inflammatory genes, which promote hypersensitivity [13,14]. The STAT signaling pathways can “cross-talk” with the MAPK pathway and the NF-κB pathway [15]. To reveal the exact molecular targets of DHM, we examined the effects of DHM on the phosphorylation of STAT5 kinases, which are the most upstream kinases responsible for mast cell activation. DHM treatment suppressed the phosphorylation levels of STAT5 in activated KU812 cells. Pimozide is an FDA-approved cell-permeable drug with STAT5-inhibitory activity [16,17]. Treatment with pimozide decreased STAT5 phosphorylation. The combination of pimozide and DHM further exaggerated the reduction in DNP-IgE-induced phosphor-STAT5. DHM synergizes with pimozide in suppressing STAT5 signaling in KU812 cells (Figure 5).

We observed a decreased phosphorylation of NF-κB and STAT, according to western blot analysis. Thus, DHM may inactivate the degranulation and cytokine production in IgE-activated KU812 cells through the suppression of the NF-κB and STAT pathways (Figure 6).

## 3. Discussion

In the present study, DHM was investigated for its ability to suppress IgE-mediated responses in cultured human mast cells. Published records have shown the anti-inflammatory properties of DHM, evidenced by the effective protection of human umbilical vein endothelial cells (HUVECs) damaged by fatty acids [18], DHM can also inhibit intracellular lipid accumulation in human acute monocytic leukemia cells (THP-1 cells) [19] and suppress the activation of the NF-κB signaling pathways and downstream pro-inflammatory cytokines in HeLa cells [13]. Our work further demonstrated that DHM counteracts the DNP-IgE-induced mediator release through its antioxidant and anti-inflammation activities in KU812 cells.

Mast cells are long-lived tissue-resident immune cells that migrate to and differentiate within tissues. In healthy tissues, mast cells are maintained in constant numbers, while the mast cell population increases dramatically in chronically allergic tissues [14]. In principle, increased mast cell numbers could be a driving mechanism in many disorders. The employed DHM did not show any cytotoxicity in KU812 cells, suggesting that these concentrations are within safety limits. DHM was found to inhibit IgE-triggered mast cell proliferation.

Previous research has demonstrated that mast cells play a crucial role in the mediation of allergic reactions via the degranulation process, predominantly caused by the antigen-IgE antibody reaction [20]. The degree of degranulation reflects mast cell activation. Tryptase is a mast cell protease that is expressed by all mast cells and contributes to inflammation in the skin and several autoimmune diseases by causing smooth muscle contraction and fibrosis [21]. The results of the present study showed that DHM has the ability to suppress the degranulation of mast cells and can be considered a pharmacological target for addressing mast cell mediator-related symptoms.

However, our study identified STAT5, but not ERKs, as involved in the mechanism integrating DHM anti-inflammation activity. IgE crosslinked with cognate receptors activates the subsequent Jak-Stat pathway in mast cells. Activated STATs bind to specific DNA sequences and initiate transcription of target genes [22]. STAT5 has been recognized in these signaling cascades that control mast cell function and expansion of the mast cell population [23,24]. The exacerbated inflammation caused by IgE was offset by the pSTAT5 inhibitor pimozide following DHM. The results of this study revealed that DHM significantly attenuated IgE-induced ROS and suppressed STAT5 phosphorylation, which could be one of the important mechanisms underlying the protective effect of DHM on abnormal mast cell proliferation and mastocytosis.

ROS-mediated oxidation of upstream kinases can influence the NF-κB, which modulates cytokine expression in innate immunity [25]. In addition, ROS also modulate STAT activation in normal and cancer cells, and in turn, STAT5 modulates ROS production, ultimately leading to the feed-forward loop that augments STAT5 activation and drives ROS formation [26,27]. These results suggest that NF-κB and ROS-mediated signaling is important in IgE-mediated mast cell activation. Our study also demonstrated that DHM mitigated the mast cell activity could help manage allergic disorders.  

## 4. Materials and Methods

The cell line identified as an immature human basophilic leukocyte KU812 cell line [11] was purchased from the Bioresource Collection and Research Center, Taiwan. Dinitrophenyl-bovine serum albumin (DNP-HSA), anti-dinitrophenyl IgE isotype (DNP-IgE), dihydromyrecitin, and pimozide were purchased from Sigma-Aldrich Chemical Co. (Merck KGaA, Darmstadt, Germany). ELISA kits for cytokine assays were purchased from eBioscience (San Diego, CA, USA). Primary antibodies against GAPDH were purchased from Santa Cruz Biotechnology Inc. (Dallas, TX, USA). Primary antibodies against phospho-IκBα, IκBα, NF-κB p65, phospho-STAT5a and STAT5 were purchased from Cell Signaling Technology, Inc. (Danvers, MA, USA). The chemical reagents used in this study were purchased from the Sigma-Aldrich Chemical Co.

### 4.1. Cell Proliferation Assay

KU812 cells were grown in a RPMI-1640 medium containing 10% heat-inactivated fetal calf serum and 1% antibiotics under standard cell culture conditions (37 °C, 5% CO_2_ in a humidified incubator). Cell viability was determined by measuring cell number. Briefly, KU812 cells were sensitized with DNP-IgE (100 ng/mL) for 16 h, stimulated with DNP-HSA antigens (100 ng/mL) for 4 h, and further treated with HSA DHM for 24 h at 37 °C. The cell numbers were counted using a hemocytometer with trypan blue staining [28].

### 4.2. Cytokines Measurement

Supernatants obtained from cell cultures in the various treatment groups were analyzed for interleukin IL-6 and tumor necrosis factor (TNF)-α using an enzyme-linked immunosorbent assay (ELISA) kit, according to the manufacturer’s instructions.

### 4.3. Reactive Oxygen Species (ROS) Measurement Using a Flow Cytometer

Flow cytometer is an useful approach to study the ROS contents in cells receiving different stimuli [29]. KU812 cells were seeded in 96-well plates at a concentration of 1 × 10^5^ cells/mL. Cells sensitized with DNP-IgE (100 ng/mL) for 16 h were incubated with DNP-HSA (100 ng/mL) for 4 h at 37 °C. The prepared cells were subsequently incubated with DHM for 24 h. After treatment with 10 μM 2, 7-dichlorofluorescein diacetate (DCFH-DA; Sigma-Aldrich, Darmstadt, Germany)) in PBS for 30 min, cells were subjected to flow cytometry (Attune NxT, Thermo Fisher Scientific, Waltham, MA, USA) analysis to examine the amount of ROS within cells.

### 4.4. Western Blot Analysis

Cells were lysed in PBS containing 1% nonidet P-40, 0.5% sodium deoxycholate, 0.1% sodium dodecyl sulfate (SDS), 5 μg/mL aprotinin, 100 μg/mL phenylmethylsulfonyl fluoride, 1 μg/mL pepstatin A, and 1 mM ethylenediaminetetraacetic acid (EDTA) at 4 °C for 20 min. Total lysates were quantified using a microBCA kit (Thermo Fisher Scientific, Waltham, MA, USA). Proteins (10 μg) were resolved by SDS-polyacrylamide gel electrophoresis and electrophoretically transferred to a polyvinylidene fluoride membrane. The membrane was blocked in 5% fat-free milk in PBST (PBS with 0.05% Tween-20), followed by overnight incubation with the following primary antibodies diluted in PBST: p65 Ab (1:1000) and pSTAT5a Ab (1:1000). The primary antibodies were removed and the membranes were washed extensively in PBST. Subsequent incubation with horseradish peroxidase-conjugated secondary antibodies (1:20,000, Santa Cruz Biotech, Dallas, TX, USA) was performed at room temperature for 2 h. The membrane was again extensively washed in PBST to remove any excess secondary antibodies, and the blots were visualized using enhanced chemiluminescence reagent (GE Healthcare, South Jakarta, Indonesia).

### 4.5. Mast Cell Degranulation Assay

A mast cell degranulation assay kit (IMM001, Millipore, MA, USA) was used to determine tryptase concentration. KU812 cells (1 × 10^6^ cells/mL) were sensitized with DNP-IgE (100 ng/mL) for 16 h. After washing twice with PBS, the cells were stimulated with DNP-HSA (100 ng/mL) for an additional 4 h. To measure tryptase release (a biomarker of degranulation) from cells, supernatants and cell lysates were incubated in a substrate solution (1.3 mg/mL of tosyl-gly-pro-lys-pNA in 0.1 M sodium carbonate) at 37 °C for 1 h, and the reaction was terminated by adding a stop solution (50 mM sodium carbonate) for 15 min at room temperature. Absorption was measured at 405 nm, and the results were expressed as the percentage of tryptase released.

### 4.6. Statistical Analysis

All data are presented as the mean ± standard deviation (SD) from three, four, or six independent experiments, as indicated. For analysis of two groups, statistical analysis was conducted using Student’s *t*-test (sigma plot 10.0. Systat Software, Inc., San Jose, CA, USA). Statistical significance was defined as *p* < 0.05, and indicated by asterisks as follows: * *p* < 0.05, ** *p* < 0.01, *** *p* < 0.001.

## 5. Conclusions

The inhibition of allergic mediator generation by mast cells is an important therapeutic strategy in the context of allergic inflammatory diseases. In the present study, we provide evidence that DHM could be developed as a mast cell stabilizer in disease therapy by suppressing IgE-dependent tryptase release, as observed by targeting STAT5 kinases, which are the upstream signaling molecules for mast cell degranulation. Moreover, DHM reduced pro-inflammatory cytokine production in a dose-dependent manner in DNP-IgEsensitized KU812 cells by suppressing the STAT5 and NF-κB signaling pathways.

## Figures and Tables

**Figure 1 molecules-26-03877-f001:**
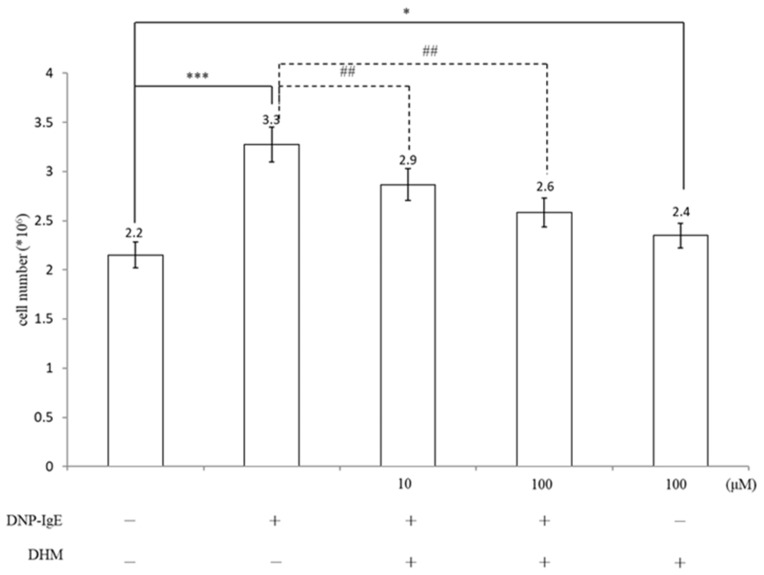
Effect of DHM on the proliferation of KU812 cells. Cell proliferation was assessed by trypan blue assay. Results show the fold relative to the control. Data represent the mean ± SD of three independent experiments. * *p* < 0.05, *** *p* < 0.001 as compared with untreated control and ## *p* < 0.01 as compared with DNP-IgE.

**Figure 2 molecules-26-03877-f002:**
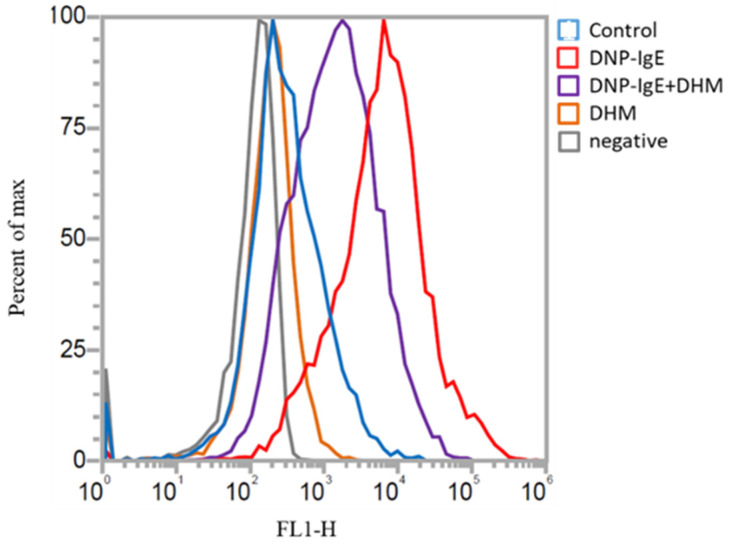
DHM (100 μM) alleviates the oxidative levels in IgE-activated KU812 cells. Representative flow cytometry profiles show changes in the levels of ROS in IgE-challenged KU812 cells after treatment with DHM. A histogram of fluorescence channel (FL1-H) versus cell count (y-axis) was generated to show cells stained with DCFH-DA in comparison to the non-sensitized control cells. Three independent, representative experiments are presented.

**Figure 3 molecules-26-03877-f003:**
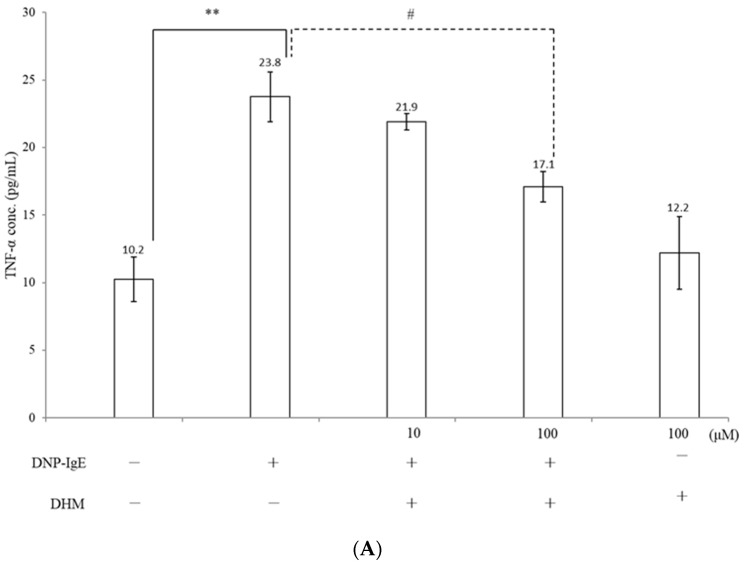
The effect of DHM on cytokine production and signaling pathway. Incubation with DHM led to decreased production of TNF-α (**A**) and IL-6 (**B**) after sensitization with DNP-IgE. All bars depict the mean of 3 repeated experiments with SD. ** *p* < 0.01, *** *p* < 0.001 as compared with the untreated control and # *p* < 0.05, ### *p* < 0.001 as compared with DNP-IgE. Levels of total and phosphorylated MAPKs (**C**) and the expression of NF-κB (**D**) were assessed by western blot.

**Figure 4 molecules-26-03877-f004:**
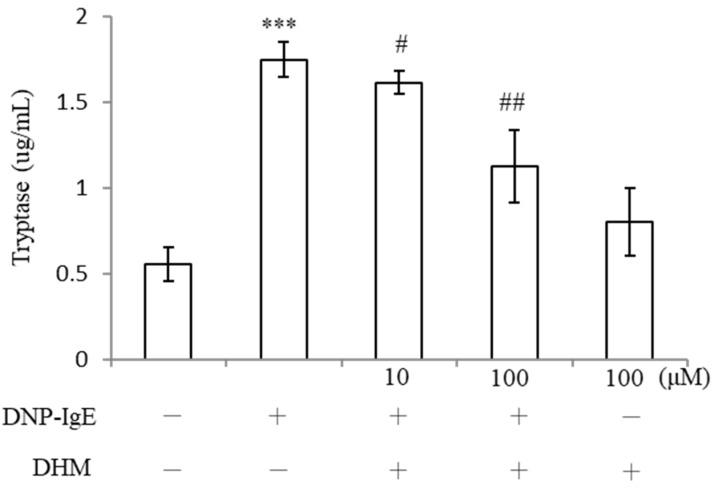
DHM reduced degranulation from KU812 cells after IgE-DNP-induced activation. KU812 cells were either sensitized with DNP-IgE or treated with DHM. The amount of tryptase from a million cells released into the supernatants was quantified using a degranulation kit after 24 h of incubation. Data represent the mean ± SD of three independent experiments. *** *p* < 0.001 as compared with untreated control and # *p* < 0.05, ## *p* < 0.01 as compared with DNP-IgE.

**Figure 5 molecules-26-03877-f005:**
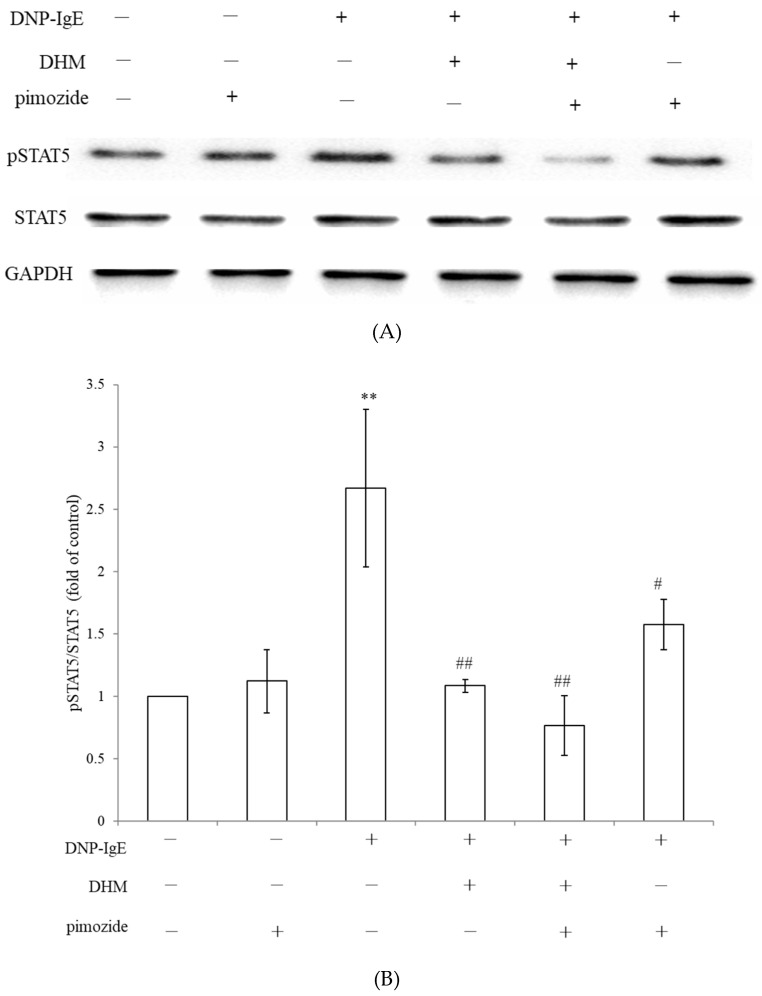
DHM inhibits the IgE-inducing STAT5 pathways in KU812 cells. Representative western blots of p-STAT5 (**A**). Protein expression in KU812 cells, cells treated with DNP-IgE, 10 µM pimozide, 100 µM DHM, or both compounds combined. Pimozide in combination with DHM suppressed the expression of p-STAT5 compared with either compound alone. The relative ratio of phosphorylated protein to total protein levels were presented as the mean ± SD (**B**). Fold of control, ** *p* < 0.01, # Fold of DNP IgE, # *p* < 0.05, ## *p* < 0.01.

**Figure 6 molecules-26-03877-f006:**
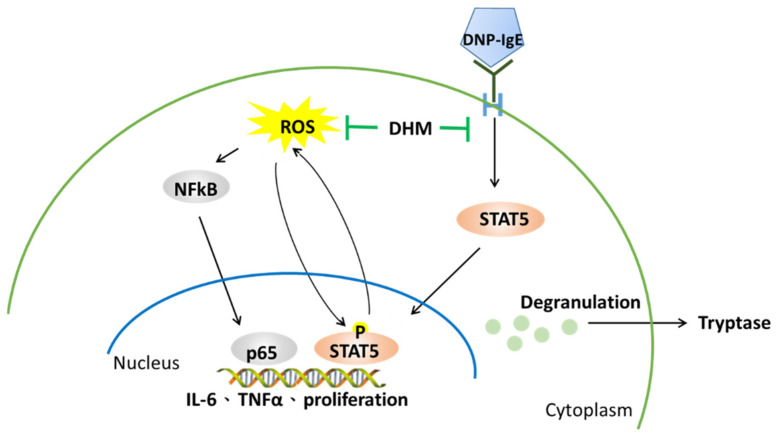
Proposed mechanism by which DHM diminishes IgE-stimulated cell proliferation, cytokine production, and degranulation in KU812 cells.

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
