# Peer review of "IgE-Induced Mast Cell Activation Is Suppressed by Dihydromyricetin through the Inhibition of NF-κB Signaling Pathway"

_molecules, 2021, doi:10.3390/molecules26133877_

Round 1

Reviewer 1 Report

The time for stimulation and sensitization keeps changing in different experiments and also from the methods section to the results sections. Please rectify the typographical error

Fig1: Are the unstimulated cells treated with DNP-BSA? If so it is hard to decipher.  Is the IgE treatment concurrent with DHM treatment?

Please find a better way to represent the cell number data by either using cell number or using percentages

Fig2: Please mention the concentration of DHM used in this experiments. Why do some experiments have DHM at 10 and 100um and this experiment does not show DHM10um

Please give the rationale behind why flow cytometry is used for ROS determination? It is not described in the methods section

Fig3: A) The errors bars for stimulated vs 10um DHM are overlapping. Can this data be reanalysed to be sure that it is significant?

B) Please show the statistics between untreated control and 100um DHM control

C) Check labelling for western blot (plus sign missing). phospho p65 is not shown. Also since these proteins would have been tested on mulitple different blots, please show densitrometric analysis for the phospho vs total protein - normalized to GAPDH. 

The conclusion that NF-kb is attenuated is not reflected in the figure.

Fig 5: Please explain in detail the rational for checking STAT5. Check the labelling for lane 4 and lane 6 above the western blot image.

Author Response

Response to Reviewer                                 June 17, 2021

Manuscript ID: molecules-1212621

Title: IgE-induced mast cell activation is suppressed by dihydromyricetin

through the inhibition of NF-kB signaling pathway

Authors: Tsong-Min Chang, Tzu-Chih Hsiao, Ting-Ya Yang, and Huey-Chun Huang *

Dear reviewers of Molecules

Thank you for reviewing our manuscript entitled “IgE-induced mast cell activation is suppressed by dihydromyricetin through the inhibition of NF-kB signaling pathway”. Our revisions in response to the reviewers’ comments are addressed below in a point-by-point manner accordingly. Many thanks again for your valuable comments and suggestions. We are looking forward to your positive decision on our article.

Response to Reviewer 1 comments:

The time for stimulation and sensitization keeps changing in different experiments and also from the methods section to the results sections. Please rectify the typographical error

Response: Line 61-63, we have carefully amended the description of stimulation and sensitization in our revised manuscript.

Fig1: Are the unstimulated cells treated with DNP-BSA? If so it is hard to decipher. Is the IgE treatment concurrent with DHM treatment?

Response: In this study the unstimulated group means cells neither been stimulated nor sensitized. The DNP-IgE label indicated cells cultured with anti-DNP-IgE for 16 h, then added DNP-HSA for 4 h. The DHM treatment indicated IgE prior DNP sensitization followed by cultured with DHM for further 24 h. Therefore, IgE treatment did not concurrent with DHM treatment.

Please find a better way to represent the cell number data by either using cell number or using percentages

Response: Line 67-68, we have replotted the y-axis in Fig. 1 which revealed the cell viability, set the cell number as vertical axis value to compare the results of various treatments.

Fig2: Please mention the concentration of DHM used in this experiments. Why do some experiments have DHM at 10 and 100um and this experiment does not show DHM10um

Response: Fig. 2 depicted the variation of intracellular ROS and the concentration of DHM used in this experiment was 100μM since this concentration of DHM exhibited stronger anti-inflammation activity.

Line 79, the first word of figure legend of Fig. 2 was amended as DHM (100 mM)

Please give the rationale behind why flow cytometry is used for ROS determination? It is not described in the methods section

Response: Line 228, in our revised version, the rationale and the reference of ROS measurement by flow cytometry [29] has been added in the Material and Method section.

Fig3: A) The errors bars for stimulated vs 10um DHM are overlapping. Can this data be reanalysed to be sure that it is significant?

Response: Line 101-105, we have amended Fig. 3A to reveal the variance between sensitized group and DHM added group.

  1. B) Please show the statistics between untreated control and 100um DHM control

Response: Line 106-108, we have amended Fig. 3B to show the statistics between untreated control and 100 uM DHM control.

  1. C) Check labelling for western blot (plus sign missing). phospho p65 is not shown. Also since these proteins would have been tested on mulitple different blots, please show densitrometric analysis for the phospho vs total protein - normalized to GAPDH. The conclusion that NF-kb is attenuated is not reflected in the figure.

Response: Line 112-114, in our revised version, the quantitative assessment of NF-κB p65 has been added as Fig.3D.

Fig 5: Please explain in detail the rational. Check the labelling for lane 4 and lane 6 above the western blot image.

Response: Line 137-145, in our revised version, the rationale for STAT5 and the references have been added into the Result section 2.5.

Line 147-149, we have also amended the labeling error of Fig.5

Reviewer 2 Report

The article by Chang et al. describes antiallergic and antiinflammatory properties of a plant-derived flavonoid dihydromirycetin (DHM). Although flavonoids and dihydromirycetin in particular are already known for such effects, the authors investigate more deeply the mechanisms of action of the compound. The article is scientifically sound and well written. Authors set up a logical sequence of experiments to track the inflammatory pathways which are affected by DHM. They also test the effect of DHM on different key inflammatory proteins. They find out that the compound is acting on the NfkB and MAPK pathways on the level of STAT5 phosphorylation or upstream. I recommend the article to be accepted for publication with certain minor refinements and/or changes, which are listed below:

1) In Paragraph 2.5 it is unclear for the reader, what the function of pimoside is. Please, describe more precisely the role of pimozide in the experiments with STAT5 phosphorylation (cite a paper describing its inhibitory function and mode of action).

2) In Fig.6 DHM is shown to block the STAT5 pathway on the level upstream of the FcεRI receptor. According to the findings of the paper it can not be concluded – it only suggests that the target lies upstream of STAT5. Please make the drawing more accurate in this regard.

3)Also, a list of putative mechanisms of DHM action upstream of STAT5 would make the discussion section stronger.

Author Response

Response to Reviewer June 17, 2021

Manuscript ID: molecules-1212621 Title: IgE-induced mast cell activation is suppressed by dihydromyricetin through the inhibition of NF-κB signaling pathway

Authors: Tsong-Min Chang, Tzu-Chih Hsiao, Ting-Ya Yang, and Huey-Chun Huang *

Dear reviewers of Molecules

Thank you for reviewing our manuscript entitled “IgE-induced mast cell activation is suppressed by dihydromyricetin through the inhibition of NF-κB signaling pathway”. Our revisions in response to the reviewers’ comments are addressed below in a point-by-point manner accordingly. Many thanks again for your valuable comments and suggestions. We are looking forward to your positive decision on our article.

Response to Reviewer 2 comments: The article by Chang et al. describes antiallergic and antiinflammatory properties of a plant-derived flavonoid dihydromirycetin (DHM). Although flavonoids and dihydromirycetin in particular are already known for such effects, the authors investigate more deeply the mechanisms of action of the compound. The article is scientifically sound and well written. Authors set up a logical sequence of experiments to track the inflammatory pathways which are affected by DHM. They also test the effect of DHM on different key inflammatory proteins. They find out that the compound is acting on the NfkB and MAPK pathways on the level of STAT5 phosphorylation or upstream. I recommend the article to be accepted for publication with certain minor refinements and/or changes, which are listed below:

1) In Paragraph 2.5 it is unclear for the reader, what the function of pimoside is. Please, describe more precisely the role of pimozide in the experiments with STAT5 phosphorylation (cite a paper describing its inhibitory function and mode of action).

1. Response: Line 141-145, in our revised version, the characters of pimozide and references have been described in the 2.5 paragraph of the Result section

2) In Fig.6 DHM is shown to block the STAT5 pathway on the level upstream of the FcεRI receptor. According to the findings of the paper it can not be concluded – it only suggests that the target lies upstream of STAT5. Please make the drawing more accurate in this regard.

2. Response: Line 161-163, we have amended the graphical summary in this revised Fig. 6.

3)Also, a list of putative mechanisms of DHM action upstream of STAT5 would make the discussion section stronger.

3. Response: Line 188-197, we have amended the Discussion section in this revised section

Reviewer 3 Report

  1. Materials and Methods Section - Unspecified source of the flavonoid Dihydromyricetin (DHM). Was it purchased pure or the result of an extraction from the Ampelopsis grossedentata?
  2. Inconsistency in the sections: Materials and Methods “Cell proliferation assay” and Results “2.1 Effect of DHM on the proliferation of KU812 cells” regarding the μg/mL and hours of use of anti-DNP-IgE and DNP-BSA in sensitizing the KU812 cells.
  3. In the section “4. DHM promotes recovery from IgE-induced degranulation” - In figure 4 the result of 80% degranulation reduction is not very evident and understandable.
  4. Section “3. Discussion” – “Our work further demonstrated that DHM counteracts the DNP-IgE-induced mediator release in a dose-dependent manner at a concentration of 10 μM”. The data shows a significant effect at 100 μM.
  5. Layout error in the Materials and Methods section - Measurement of reactive oxygen species (ROS).
  6. References Section - Reference 1 and Reference 6 concerns the same citation
  7. Inappropriate use of the term antagonist in the Abstract section, last sentence. There is no evidence that DHM binds to the receptor and blocks signal transduction

Author Response

Response to Reviewer                                 June 17, 2021

Manuscript ID: molecules-1212621

Title: IgE-induced mast cell activation is suppressed by dihydromyricetin

through the inhibition of NF-kB signaling pathway

Authors: Tsong-Min Chang, Tzu-Chih Hsiao, Ting-Ya Yang, and Huey-Chun Huang *

Dear reviewers of Molecules

Thank you for reviewing our manuscript entitled “IgE-induced mast cell activation is suppressed by dihydromyricetin through the inhibition of NF-kB signaling pathway”. Our revisions in response to the reviewers’ comments are addressed below in a point-by-point manner accordingly. Many thanks again for your valuable comments and suggestions. We are looking forward to your positive decision on our article.

Response to Reviewer 3 comments:

Materials and Methods Section - Unspecified source of the flavonoid Dihydromyricetin (DHM). Was it purchased pure or the result of an extraction from the Ampelopsis grossedentata?

1.Response: Line 213, the pure DHM was purchased from Sigma-Aldrich chemical company and has been amended in the Material and Methods section.

2.Inconsistency in the sections: Materials and Methods “Cell proliferation assay” and Results “2.1 Effect of DHM on the proliferation of KU812 cells” regarding the μg/mL and hours of use of anti-DNP-IgE and DNP-BSA in sensitizing the KU812 cells.

2.Response: Line 221-223, Materials and Methods “Cell proliferation assay”; Line 61-63, Results “2.1 Effect of DHM on the proliferation of KU812 cells”, we have carefully amended the description of stimulation and sensitization in our revised manuscript.

3.In the section “4. DHM promotes recovery from IgE-induced degranulation” - In figure 4 the result of 80% degranulation reduction is not very evident and understandable.

3.Response: Line 123-126, we have carefully amended the description the result of degranulation in Result 2.4 section as followings: The addition of 100 mM DHM reduced the DNP-IgE induced release of tryptase (1.75 ± 0.1 mg/mL) for DNP-IgE group to (1.13 ± 0.211 mg/mL) for DNP-IgE plus DHM group), which was nearly 64.5% of the value of the DNP-IgE plus group. (1.13/1.75 ´ 100% = 64.5%)

4.Section “3. Discussion” – “Our work further demonstrated that DHM counteracts the DNP-IgE-induced mediator release in a dose-dependent manner at a concentration of 10 μM”. The data shows a significant effect at 100 μM.

4.Response: Line 170-172, we have carefully amended the description in the Discussion section

5.Layout error in the Materials and Methods section - Measurement of reactive oxygen species (ROS).

5.Response: Line 227-235, we have carefully amended the description of the ROS section

6.References Section - Reference 1 and Reference 6 concerns the same citation

6.Response: We have checked and amended the references of this revised section.

7.Inappropriate use of the term antagonist in the Abstract section, last sentence. There is no evidence that DHM binds to the receptor and blocks signal transduction

7.Response: Line 21, we have replaced the “antagonist” with “mitigate” of this revised section.

Round 2

Reviewer 1 Report

Revised comments are satisfactory

Reviewer 2 Report

Dear Authors!

Thank you for improving thr manuscript according to my comments.

It is now ready for publication. 

Reviewer 3 Report

No comment